# Plant GDSL Esterases/Lipases: Evolutionary, Physiological and Molecular Functions in Plant Development

**DOI:** 10.3390/plants11040468

**Published:** 2022-02-09

**Authors:** Gaodian Shen, Wenli Sun, Zican Chen, Lei Shi, Jun Hong, Jianxin Shi

**Affiliations:** Joint International Research Laboratory of Metabolic and Developmental Sciences, School of Life Sciences and Biotechnology, Shanghai Jiao Tong University, Shanghai 200240, China; shengaodian@sjtu.edu.cn (G.S.); sunwenli@sjtu.edu.cn (W.S.); czcqbqmx@163.com (Z.C.); shileilei@sjtu.edu.cn (L.S.); m13605151497@163.com (J.H.)

**Keywords:** lipases, lipid metabolism, plant-environment interactions, reproductive development, vegetative development

## Abstract

GDSL esterases/lipases (GELPs), present throughout all living organisms, have been a very attractive research subject in plant science due mainly to constantly emerging properties and functions in plant growth and development under both normal and stressful conditions. This review summarizes the advances in research on plant GELPs in several model plants and crops, including Arabidopsis, rice, maize and tomato, while focusing on the roles of GELPs in regulating plant development and plant–environment interactions. In addition, the possible regulatory network and mechanisms of GELPs have been discussed.

## 1. Introduction

GDSL esterase/lipases (GELPs) represent a variety of lipolytic enzymes that hydrolyze diverse lipidic substrates including thioesters, aryl esters, and phospholipids [1]. GELPs contain a unique and conserved GDSL motif GDSxxDxG at their N-terminus that distinguish them from classic lipolytic enzymes that harbor a conserved motif GxSxG [1]. GELPs are widely distributed in microbes, animals and plants, playing vital roles in growth, development and stress responses [2,3].

Compared with bacteria and animals, plants retain substantial amounts of GELPs that are composed of three subfamilies [2,3], indicating that GELPs have acquired essential physiological functions in plants. To date, genome wide identification of GELPs has been carried out in various plant species (Table 1), including *Arabidopsis thaliana* [4], *Oryza sativa* [5], *Brassica rapa* [6], *Vitis vinifera* [7], *Zea mays* [8], *Glycine max* [2,9], and Rosaceae (such as *Fragaria vesca*, *Prunus persica*, *P. avium*, *P. mume*, Pyrus *bretschneideri*, and *Malus domestica*) [10]. However, the enzymatic or biological functions of most of GELPs have not been characterized *in planta* [2,3].

The classical lipolytic enzymes have a catalytic Ser residue while GELPs have four invariant catalytic residues—Ser, Gly, Asn, and His—that play important roles in enzyme catalysis [1]. Therefore, GDSLs are also known as SGNH lipases. Unlike other lipolytic enzymes, GELPs are more flexible in structure with more flexible active sites, broadly diverse substrates and multifunctional properties [1,2,3,4,5]. In addition, while other lipolytic enzymes hydrolyze lipids, GELPs can hydrolyze sulfur, aryl vinegar and amino acids, in addition to lipids [1]. Understanding physiological and molecular functionalities of plant GDSL lipases will not only facilitate functional characterization of these fascinating lipolytic enzymes, but also the possible application in plant breeding for resilient crops with multiple resistant/tolerant traits to environmental changes, to secure food supply under changing climate conditions. In this review, we have focused on new, recent information on plant GELPs and their roles in plant growth and development.

## 2. Roles of GELP in Plant Land Colonization

One important evolutionary feature of plant land colonization was the formation of the cuticle approximately 450 million years ago [11], which was together with the formation of a cell wall before the first colonization of true land plants [12]. Cuticle formation, particularly cutin and suberin, involves many proteins and enzymes including GELPs that play important roles in the polymerization. GELPs are found across all land plants [13], including anciently conserved spermatophytes and bryophytes such as *Selaginella moellendorffii* and *Physcomitrella patens* [14]. In addition, GELPs emerged as early as in *Penium margaritaceum* charophytes, the last common ancestor of the Zygnematophyceae and land plants [15]. These results highlight important roles of GELPs in plant land colonization. In addition, GELPs expand significantly in an ancient lycophyte (*Selaginella moellendorffi*) that diverged shortly after land plants had evolved vascular tissues, which also indicates a role for GELPs in the early evolution of vascular plants [3]. Evolutionarily, the emergence of GELPs in charophytes is supposed to contribute to plant land colonization by providing biological services including the conservation of water in a desiccating environment [11] and biological interactions between cuticle and other cell wall components [12]. Nevertheless, more comparative genomic and molecular studies are needed to better understand the exact roles of GELPs in plant land colonization.

## 3. Roles of GELPs in Plant Development and Plant Metabolism

GELPs have been suggested to play crucial roles in almost all aspects of plant growth and development [2,4,9,16]. Increasing studies confirmed their functions in both vegetative and reproductive development, and plant metabolism as well. The exploration of metabolic and physiological functions of GELPs will provide new routes for redesigning the regulatory networks of plant growth and development.

### 3.1. Vegetative Development

#### 3.1.1. Seed Germination

In germinating seeds, the hydrolysis of stored lipids by lipases plays a crucial role in the initial stage of seed germination. A study investigated the expression of 110 *GELP* genes over three time points during rice seed germination, and found that most of them are expressed during germination, and about one third of them are highly expressed across all three germination stages [17], indicating an important role for GELPs in seed germination. Nevertheless, none of them have been experimentally validated. In *Brassica napus*, BnSCE3/BnLIP2, a sinapine esterase, catalyzed the hydrolysis of sinapine during early seed germination, and overexpression of *BnSCE3/BnLIP2* slightly enhanced seed germination rate and seedling development [18]; however, its function in seed germination has not been confirmed using loss-of-function mutant. In Arabidopsis, *LIP1*, whose expression is highly induced by GA and repressed by DELLAs during seed imbibition, controls embryo axis elongation before germination, implying its involvement in seed germination [19]; however, no functional analysis has been performed. The function of GELPs in seed germination remains to be determined in more plant species.

#### 3.1.2. Coleoptile Elongation

Plant coleoptile elongation is influenced by various genetic and environmental factors. The regulatory mechanisms underlying this process and particularly the involvement of GELP are poorly understood. Rice *GER1* (*GDSL containing enzyme rice 1*) is the only *GELP* that has been reported in this area. GER1 was inducible by jasmonic acid (JA) and light (both red and far-red light), and negatively regulated coleoptile elongation in rice [20].

#### 3.1.3. Lateral Root Development

Suberin plasticity is highly associated with plant lateral root development, and GELPs have recently been found to be involved this auxin signaling-mediated process, in which different sets of auxin responsive *GELPs* modulate both suberine deposition and degradation in the cell walls of endodermal cells overlying the lateral root primordium to facilitate the lateral root development [21]. Auxin repressible *GELP22*, *GELP38*, *GELP49*, *GELP51* and *GELP96* were needed for suberin polymerization in the apoplast while auxin inducible *GELP72*, *GELP73* and *GELP81* were required for suberin degradation during lateral root emergence [21]. Single loss-of-function mutants of five suberin-biosynthetic GELPs did not show defectiveness in lateral root development, while lateral root emergences in triple mutants of suberin-degrading GELPs were all delayed [21]. Additionally, loss-of-function mutants of *GLIP2* were more sensitive to auxin with increased expression of auxin signaling genes, such as *IAA1* and *IAA2*, and increased numbers of lateral roots [22]. Notably, although Arabidopsis *CDEF1* was expressed in the zone of lateral root emergence, loss-of-function mutants of *CDEF1* did not show any defects in lateral root emergence [23], likely due to redundancy among *GELPs*. The involvement of GELPs in lateral root development in other plants remains unknown.

#### 3.1.4. Root Development

Exogenous ethylene restricts root growth. A recent study provided novel mechanistic insights into ethylene signaling in rice, in which the ER localized GELP MHZ11 (mao huzi 11) was necessary for ethylene signal transduction, and the overexpression of *MHZ11* resulted in shorter roots in the presence of air or ethylene [24]. Loss-of-function of *MHZ11* did not alter coleoptile and root growth in the presence of air under dark, but significantly altered the ethylene sensitivity in rice primary roots, in which wild type root growth was drastically repressed while *mhz11* root growth was barely inhibited [24]. In response to ethylene, MHZ11 lipase inhibited *OsCTR2* phosphorylation and ethylene signaling by maintaining low sterol levels on the ER membrane [24], thus activating ethylene response in root growth. The detailed network of GELPs in the ethylene signal-mediated root development merits further investigations.

#### 3.1.5. Plant Height

The epidermis is one of the multiple regulatory factors of plant height, an important agronomic trait [25]. Theoretically, epidermal apoplast localized GELPs are expected to influence plant height. Indeed, DR (drooping leaf), a rice GELP without localization information, and WDL1 (wilted dwarf and lethal 1), an ER localized Arabidopsis GELP, were reported to affect plant height. *DR* was required for leaf silica deposition; *dr* mutant exhibited drooping leaves with small midrib, shorter panicle, and reduced plant height [26]. Notably, silica is deposited in the epidermis just beneath the cuticle layer. *WDL1* was involved in cutin patterning in the epidermis; *wdl1* mutant was dwarf and died soon after the seedling establishment due to over loss of water [26]. These studies indicated that GELP-mediated cuticle and silica metabolism affects plant height, although the exact mechanisms have not been resolved.

#### 3.1.6. Stomata Development

Epidermal cell patterning is closely associated with stomata development and plays key roles in plant–environment interactions. Two studies demonstrated the involvement of GELPs in stomata development. Arabidopsis *wdl1* mutant exhibited smaller stomatal and pavement cells but more stomatal and pavement cells per unit area [27]. Arabidopsis OSP1 (occlusion of stomatal pore 1), an ER and lipid droplet localized GELP, was reported to be involved in the early wax biosynthesis and stomatal cuticular ledge formation [28]. Almost half of the mature stomata in *osp1* were occluded with reduced stomata conductance, due to the fully covered cuticular layer formed between the edges of two guard cells surrounding the pore [28]. It seems that disrupting *GELP* mediated-epidermal cell patterning affects stomata development.

### 3.2. Reproductive Development

#### 3.2.1. Flower Development

Up to date, several GELPs have been reported to be involved in flower development. DAD1 (defective in anther dehiscence1) in Arabidopsis was reported to catalyze the initial step of JA biosynthesis and coordinates JA-dependent flower development [29]. *dad1* mutant was defective in anther dehiscence, pollen maturation, and flower opening [29]. *CUS2* (*cutin synthase 2*), an orthologue of tomato *CD1/CUS1* [14,30] in Arabidopsis was found to be not required for initial formation but required for maintenance of cuticular ridge on sepal epidermis [31]. At early stages, *cus2* mutant sepals showed similar cuticular ridge distributions and patterns similar to WT sepals. At mature stage, *cus2* mutant sepal epidermis lacked cuticular ridges. Thus, mutant flowers were found to be more permeable to toluidine blue dye although the morphology of sepals and petals were normal [31]. Our previous study found that expression levels of two Arabidopsis GELP genes, *GELP15* and *GELP39*, are significantly down-regulated in *hsp70-16* mutant sepals grown under normal and mild heat stress conditions [32], which was found to be concomitant with fused lateral sepals and abnormal flower opening [33].

#### 3.2.2. Anther and Pollen Development

Lipid metabolism is critical for the building of the anther cuticle and the pollen wall, and mutants of genes associated with lipid metabolism often show male sterility [34,35]. It is reported that, about 10% of predicted oilseed rape GELPs are specifically expressed in fertile but not in male-sterile buds, implying their functions in pollen development [6]. Mutants of some *GELPs* displayed similar male sterility phenotypes in different plant species, indicating conservative roles for *GELPs* in plant male fertility.

Besides *DAD1*, which functioned in anther dehiscence and pollen maturation, more *GELPs* have been reported to be involved in anther and pollen development in Arabidopsis. *RVMS* (*reversible male sterile*) was found to encode a GELP with lipase activity that localizes especially in microspore mother cells, meiotic cells and microspores, and participates in pollen development [36]. *rvms* mutant displayed defective nexine formation, degenerated microspore cytoplasm, and complete male sterility [36]. Notably, *rvms* was fertile at low temperatures [36]. In addition, *GELP77* was reported to be necessary for pollen fertility and pollen dissociation. *gelp77* pollen grains lacked well-organized reticulate surface structure; they were shrunken and stuck to each other that could not be released from mature anthers [37]. Furthermore, a pollen coat protein encoding the gene *EXL6* (extracellular lipase 6), the target of a key Arabidopsis pollen development regulator AMS (aborted microspores) [38], was reported to be highly expressed at stage corresponding to microspore pollen development. Knockdown mutants of *EXL6* exhibited defective pollen grains due to cytoplasmic degradation after the tetrad stage [6].

In rice, *RMS2* (*rice male sterility 2*) encodes an ER-localized GELP with esterase activity that is required for male fertility. *rms2* displayed complete male sterility with defective middle layer and tapetum degradation, abnormal cuticle and exine formation, and irregular central vacuole development [39]. In addition, the ER localized OSGELP34 [40,41] and peroxisome localized OsGELP110/OsGELP115 [40] were found to catalyze different compounds for pollen exine development. *OsGELP34* was primarily expressed in anthers during pollen exine formation; *osgelp34* mutant showed abnormal exine and altered expression of many key pollen development genes, their anthers were light yellow and smaller with nonviable pollens [40,41]. *OsGELP110* is the most homologous to *OsGELP115*; both are dominantly expressed in anthers with similar patterns. The single mutant of either *OsGELP110* or *OsGELP115* did not affect male fertility, but double mutants were male sterile, harboring smaller and light yellow anther with aborted pollen grains [40]. Notably, compared with *osgelp34,* anthers in *osgelp110osgelp115* were more prominent with un-degenerated swollen tapetum and endothecium layer, and the foot layer in *osgelp110osgelp115* mutant was continuous without obvious intermission [40].

In maize, *IPE2* (*irregular pollen exine 2*), encoding an ER localized GELP, is specifically expressed in developing anthers. *ipe2* mutants, exhibiting irregular anther cuticle and pollen nexine, delayed degeneration of tapetum and middle layer, were male sterile [42]. *ZmMs30* encodes an ER localized GELP, and is specifically expressed in maize anthers similar to IPE2. *ms30* mutant exhibited a defective anther cuticle, an irregular foot layer of pollen exine, and complete male sterility [8]. Importantly, male sterility in *ms30* was stable in different inbred lines with diverse genetic backgrounds, without any penalty on maize heterosis and production, providing a valuable male-sterility system for the hybrid breeding maize [8]. Notably, *IPE2* phylogenetically is distant from *MS30* [42].

#### 3.2.3. Pollen-Stigma Interaction and Pollinator Attraction

For successful pollination, the stigma cuticle has to be destroyed by lipases-like cutinase. The identification of such plant cutinases is indispensable for elucidating the molecular mechanisms of pollen–stigma interaction. Arabidopsis *CDEF1* was found to be expressed in pollen and pollen tubes, and to be associated with pollen-stigma interaction [23], however, its function *in planta* remains unknown. Pollination begins with the hydration in which lipids and proteins are important. Arabidopsis *EXL4* alone was reported to be required for the hydration on the stigma, the initial step of pollination. *EXL4* also functioned together with *GRP17* to promote the initiation of hydration. *exl4* mutant exhibited a delayed pollen hydration [43].

Attraction of pollinators is a unique feature for pollination in plants. A study in the tropical tree *Jacaranda mimosifolia* demonstrated a role for GELPs in this process. JNP1 (jacaranda nectar protein 1) is an extracellular GELP protein present in the nectar of *J. mimosifolia* that displays lipase/esterase activities. Because of the fact that *J. mimosifolia* nectar contains lipophilic particles and accumulates large amount of free fatty acids within the nectar, JNP1 was supposed to release fatty acids from lipid particles to attract pollinators [44]. In addition, a tomato GELP, Sl-LIP8, was reported to be involved in the production of fatty acid-derived organic compounds [45]. Based on the above reports, GELPs are supposed to be functional in pollinator attraction.

#### 3.2.4. Fruit Development

A bioinformatics study of GELPs in Rosaceae genomes indicated that a set of GELPs likely participated in fruit development [10]. Nevertheless, except for increasing the glossy of the fruit surface, the mutation of *GDSL1* [46], *SlGDSL2* [47], *CUS1/CD1* [14,30] in tomato did not affect fruit size. The involvement of GELPs in the development of other fruits is rarely reported.

#### 3.2.5. Seed Development

Seed development is promoted significantly in transgenic plants overexpressing the *GLIP* genes. Expression of a *Brassica napus GLIP* gene *BnSCE3/BnLIP2* under a seed-specific promoter in oilseed rape resulted in enhanced weight and size of the transgenic seeds [18]. Ectopic expressing of a cotton *GhGLIP* in Arabidopsis increased seed size and weight [48]. Arabidopsis *RGE1* (*retarded growth of embryo 1*) is a nucleus localized bHLH transcription factor that controls embryo development from the endosperm; *rge* mutants displayed smaller and shriveled seeds, in which the expression of two *GELP* genes was especially suppressed [49]. However, the single mutant of either of *GELP* gene did not show any defectiveness in seeds due to possible functional redundancy [49].

### 3.3. Plant Metabolism

#### 3.3.1. Lipid Metabolism

GELPs possess extensive hydrolytic activity, such as thioesterase, protease, phospholipase, and arylesterase activity, making the identification of their substrates and biochemical functions in vivo a great challenge, especially in plants. Generally, their primary functions are predicted to be related to lipid metabolism to produce storage oil in seeds, cuticular lipids to cover and decorate organ surfaces, oxylipins, and other signaling molecules [1,2,3,9,10,11]. Indeed, lipolytic activity of plant GELPs has been confirmed using p-nitrophenyl butyrate as a common substrate. Those GELPs include CaGLIP1 [50], SFAR4 (seed fatty acid reducer 4) [51], GLIP2 [22], CDEF1 [23], OsGLIP1 [52], OsGLIP2 [52], ZmMs30 [8], EXL4 [43], JNP1 [44], and RMS2 [39]. Among them, cell wall or aqueous apoplast localized GELPs often acted as esterases or hydrolases, such as CDEF1 (cuticle destructing factor 1) in Arabidopsis and CUS1 in tomato. CDEF1 functioned as a cutinase disrupting pollen cuticle [23] while CUS1 was reported to be essential for cutin deposition on fruit epidermis via catalyzing the esterification of both primary and secondary alcohol groups [30]. In addition, PLIP1 (plastid lipase 1), a plastid phospholipase, contributed to seed oil biosynthesis; *plip1* mutant seeds contain 10% less oil while overexpressive mutant seeds contain 40–50% more seed oils [53].

#### 3.3.2. Cell Wall Metabolism

So far, only three GELPs have been reported to be involved in cell wall patterning. The first two are Golgi-localized GELPs in rice [54,55]. Rice BS1 (brittle leaf sheath1) deacetylated hemicellulose xylan and maintains acetylation homeostasis of the xylan backbone that is vital for secondary cell wall development. *bs1* mutants showed reduced plant height, smaller panicle, reduced tiller, and significantly decreased yield and grain weight [55]. Rice DARX1 (deacetylase on arabinosyl sidechain of xylan1) deacetylated arabinosyl residues of xylan and modulates the arabinoxylan acetylation profiles that is important for secondary wall formation. *darx1* mutants showed lower contents of cellulose, disrupted secondary wall formation and patterning, reduced mechanical strength such as easily broken internodes and drooping leaves, and slightly decreased plant height [54]. The third one is a cotton *GhGDSL*, which exhibited secondary cell wall stage-specific expression during cotton fiber development, and a 194 bp region in the 5′ of *GhGDSL* controls its expression, playing an important role in the secondary cell wall formation during fiber development [56].

#### 3.3.3. Secondary Metabolism

However, many GELPs can catalyze other reactions, showing additional transferase activities. Extracellular space localized GELPs often acted as acyltransferases, for example, SlCD1 (cutin defective 1) [57] GDSL1 in tomato [46]. CD1 catalyzed polymerization of cutin by transferring the hydroxyacyl group from its substrate 2-mono(10,16-dihydroxyhexadecanoyl)glycerol to the growing cutin polymer, GDSL1 catalyzed the multiple crosslinks present in the cuticular matrix via this transesterification activity. A novel allelic mutant of *SlCD1*, *slgdsl2*, lacked intact fruit epidermal cutin, confirming the role for *SlCD1/SlGDSL2* in cutin polymerization [47]. In addition, TcGLIP in *Tanacetum cinerariifolium* harbored transferase activity in vivo and esterase activity in vitro, which is involved in pyrethrins biosynthesis [58]. Tomato SlCGT (chlorogenate: glucarate caffeoyltransferase), the first GELP that lost hydrolytic activity but acquired acyltransferase activity, was found to participate in the synthesis of hydroxycinnamate esters by employing amino acid residues [59]. Another tomato GELP Sl-LIP8 cleaved glycerolipids to produce short chain fatty acid derived volatile organic compounds (FA-VOCs), *sl-lip8* mutant exhibited significantly reduced contents of several C5 and C6 FA-VOCs [45]. Supportively, comparative metabolomics studies between wild type and *rms2* mutant anthers demonstrated that, in addition to lipid metabolism, *RMS2* functions in other metabolic processes such as plant hormone biosynthesis and signaling and secondary metabolism [39].

Interestingly, a recent study revealed that *dr*—another allelic mutation of *BS1* in the Nipponbare background—lacks an intact silica layer in its leaf epidermis [26], implying a novel function of *GELP* in silica accumulation to maintain erect leaf morphology in rice. Nevertheless, *bs1* [55] and *dr* [26] showed different phenotypes, indicating that functions of the same GELP are controlled by different alleles and different genomic backgrounds, which merit further validations. In addition, an apoplast localized GELP in wheat, XAT (xanthophyll acyltransferase), was reported to catalyze the esterification of lutein for the synthesis of xanthophyll esters. Notably, XAT possesses esterase and lipase activity [60].

## 4. Roles of GELPs in Plant-Environment Interactions

Extensive studies have demonstrated that multiple GELPs are inducible by various abiotic and biotic stresses in some species, suggesting their defensive roles in both biotic and abiotic stresses [9,17]. Elucidation of precise roles for GELPs in synchronizing plant responses to both abiotic and biotic stresses would benefit breeding for more resilient crops to deal with changing climates.

### 4.1. Plant-Abiotic Stress Interaction

Rapid water loss is one of the important features of cuticle defective mutants; it is particularly true in mutants of GELPs with cutinase or transferase activity. Compared with wild type, mutants of rice *WDL1* [27] and tomato *GDSL1* [46], *SlGDSL2/CD1* [47,57] all showed an increased transpiration rate. Nevertheless, the water permeance of the isolated enzyme-treated cuticle from *slcus1* was found to be similar to wild type [30]; suggesting that the levels of cutin cross-linking do not affect water permeance, in agreement with reports that waxes are major determinants of cuticle permeability. On the other hand, *osp1* mutant in Arabidopsis displayed lower transpiration rate and enhanced drought tolerance [28]. In addition, overexpression plants of *CaGLIP1* [50] and *GmGELP28* [9] were highly tolerant to drought; their exact functions in drought tolerance are not yet confirmed genetically.

Several GELPs are associated with salt stress or ABA-mediated abiotic stress. *gelp^quint^* suberin-biosynthetic mutants [21] were more sensitive to salt stress while *GmGELP28* overexpression [9] plants were highly tolerant to salt. Both *PLIP2* and *PLIP3* contributed to ABA-induced JA accumulation, participating in ABA-mediated abiotic stress responses [61], while overexpression of *SFAR4* [51] and *CaGLIP1* [50] enhanced tolerance to osmotic and oxidative stress, respectively.

Nevertheless, the involvement of GELPs in plant–abiotic stress interaction is obtained mainly from bioinformatics prediction and overexpression studies. More and more evidence, particularly regarding the mechanisms behind and from mutant analyses are needed for potentially breeding abiotic tolerant plants via genetic engineering.

### 4.2. Plant-Biotic Stress Interaction

Most plant–biotic stress interaction studies have been carried out in Arabidopsis and rice with mutant analyses, which indicate an important role of GELPs-mediated lipid metabolism and hormone signaling in plant immunity.

In Arabidopsis, GELPs are essential for ethylene-associated systemic resistance. The SA-inducible *GLIP1* was known to be involved in the plant resistance to necrotrophic pathogen, the fungus (*Alternaria brassicicola*). *glip1* plants were significantly susceptible to pathogen infection [62]. *GLIP1* regulated resistance by direct disruption of the structure of fungal spore cell wall or membrane [62], and by positively and negatively feedback regulation of ethylene signaling [62,63]. In addition, *GLIP2* was reported to be inducible by JA, SA and ethylene; *glip2* mutants were more sensitive to auxin and more susceptible to necrotrophic bacteria *Erwinia carotovora* [22]. *GLIP2* was found to function in pathogenic responses through down-regulation of auxin signaling [22]. In rice, *OsGLIP1* and *OsGLIP2* negatively regulated plant’s resistance to both bacterial and fungal pathogens via modulating lipid metabolism; simultaneous down-regulation of both of them enhanced the resistance to both bacterial and fungal pathogens while the overexpression of either of them compromises tolerance [52]. In addition, expressions of 5 *GELP* genes in rice infected with *Fusarium verticillioides* was accompanied with impaired mobilization of lipids [17].

Furthermore, sporadic relevant reports could be found in pepper and sunflower; however, none of them has been confirmed with loss-of-function mutants. Transcripts of *CaGLIP1* accumulated in response to multiple defense hormones, such as SA, JA, and ethylene, and to infection with virulent and avirulent strains of *Xanthomonas campestris pv. vesicatoria*, while *CaGLIP1* overexpression plants were hypersensitive to infection of *Pseudomonas. syringae pv. tomato* [50]. In sunflower, a pan-genome analysis indicated the contribution of introgressions from wild species to downy mildew resistance, in which a GELP sits within the genomic region associated with resistance [64].

## 5. Possible Regulatory Factors and Networks

Very few studies have explored the regulatory aspects of GELPs, although bioinformatics predication of putative cis-elements has been done in many plant species [5,9,19,48]. It seems from the available information that GELPs are direct targets of multiple transcription factors and are essential elements in the signaling pathways of several phytohormones. Unearthing such regulatory networks is of importance in better understanding GELPs for potential application in breeding.

So far, several transcription factors have been reported to regulate the expression of *GELPs* directly or indirectly. First, SHN transcription factors were known regulators of cuticle deposition [65,66]. In Arabidopsis, *GELP60* and *GELP95* were targeted by SHN for cutin deposition and flower organ surface patterning [66]. In tomato, expression levels of *SlGDSL1* and *SlGDSLa* were downregulated in tomato *SHN3RNAi* mutant [67]. Second, ATML1 and PDF2, two epidermis specification regulators, targeted Arabidopsis *LIP1* at the L1 box promoter sequence that is conserved in promoter regions of epidermis-specific genes. *LIP1* expressed specifically in the epidermis and its L1 box sequence mediated GA-induced transcription in seed germination [19]. Third, UDT1 (Undeveloped Tapetum1) [68] and PTC1 (Persistent Tapetal Cell1) [69], two important early anther development regulators in rice, both targeted and regulated expression of *RMS2* [70], regulating early anther development. Fourth, MYB transcription factors were known to play important roles in almost all aspects of plant development. A cotton MYB transcription factor, GhMYB1, targeted *GhGDSL* at the MYB1AT sequence, regulating the expression of *GhGDSL* during fiber development [56].

Other factors have also been found to regulate *GELP* during plant development and response to the environment. Arabidopsis *OSP1* acted upstream of *MAH1* or *CER1* to regulate epidermal permeability and OCL formation and drought tolerance [28]. Furthermore, Arabidopsis *GELP15* and *GELP39*, two sepal expressed *GELPs*, participated in a *HSP70-16* mediated transcription regulatory network that maintains metabolic homeostasis for flower opening [32].

Regarding signaling pathways, besides the involvement of Arabidopsis *LIP1* in GA signaling, rice *MHZ11* was found to be involved in ethylene signaling, which acts upstream of *OsCTR2* in the ethylene signaling pathway, reducing sterol levels to disrupt receptor-*OsCTR2* interactions and *OsCTR2* phosphorylation for initiating ethylene signaling, thus regulating root growth [24]. Recently, two studies highlighted the participation of GELPs in plant immunity signaling. When overexpressing *Hrip1*, an elicitor isolated from necrotrophic fungus *Alternaria tenuissima,* the expression levels of two *GELPs* (*GLIP1* and *GLIP4*) were upregulated, which was associated with enhanced resistance to insect *Spodoptera exigua* [71], indicating the involvement of *GELPs* in *Hrip1* triggered insect resistant responses. In addition, two pathogen-responsive MAPKs, MPK3 and MPK6, regulated expression of *GLIP1*, *GLIP3* and *GLIP4*, contributing resistance to *Botrytis cinerea*, independent of JA and ethylene [72].

## 6. Concluding Remarks and Future Prospects

The GELP family is diverse but conserved in land plants; their isozymes differ in activity, substrate specificity, and product formation. GELPs and their hydrolyzing products play multiple roles in plant growth and development and responses to biotic and abiotic stresses. Our review provides contemporary information for the plant GELPs (Figure 1). Over the past two decades, bioinformatics has predicted the presence of GELPs in various plants including several important crop plants (Table 1), and a number of GELPs have been successfully characterized, primarily in Arabidopsis, rice and tomato (Table 2). Even with this, our understanding regarding their functions in plants, the mechanisms underlined and the regulatory aspects, is unable to meet the need for maintaining sustainable agriculture in changing climates. Natural variations in GELP genes provide valuable genetic resources for better understanding their physiological and evolutional importance in plant adaption. To better understand the exact roles of GELPs in plant development and stress response and to facilitate breeding for resilient crops to effectively secure food supply, combined genomics with genetic, biochemical and molecular approaches are needed.

## Figures and Tables

**Figure 1 plants-11-00468-f001:**
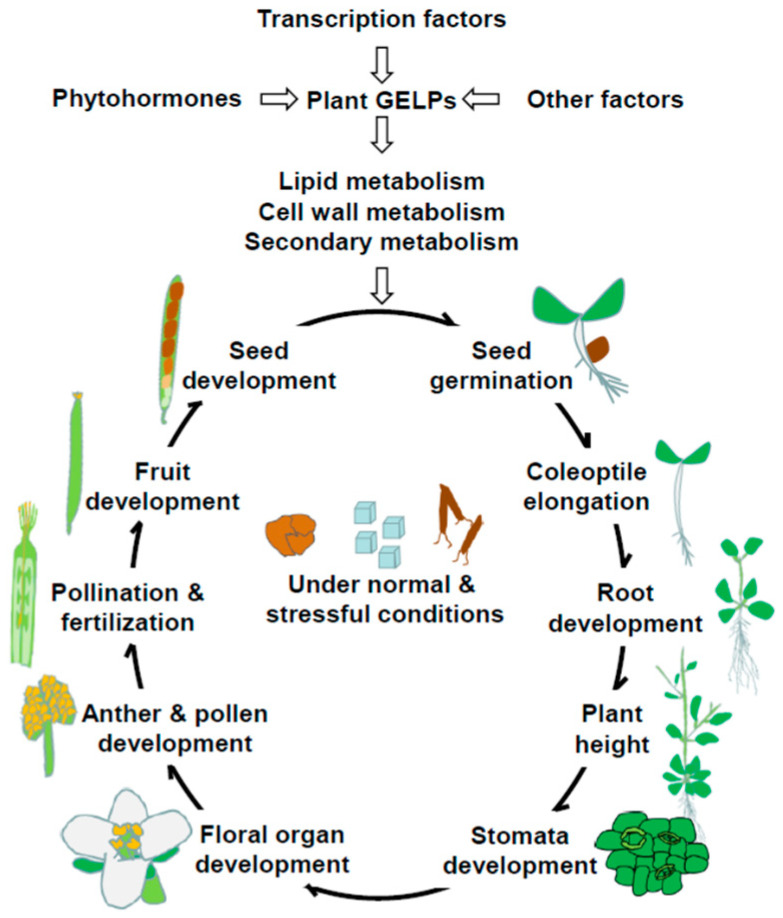
Roles and possible regulatory networks for plant GELPs in plant development and plant-environment interaction.

**Table 1 plants-11-00468-t001:** Numbers of plant GELPs predicted through whole genome identification.

Species	Numbers	Reference
*Arabidopsis thaliana*	105	[4]
*Brassica rapa*	121	[6]
*Fragaria vesca*	96	[10]
*Glycine max*	194	[2,9]
*Malus domestica*	150	[10]
*Oryza sativa*	114	[5]
*Prunus avium*	70	[10]
*Prunus mume*	90	[10]
*Prunus persica*	97	[10]
*Pyrus bretschneideri*	94	[10]
*Vitis vinifera*	83	[7]
*Zea mays*	103	[8]

**Table 2 plants-11-00468-t002:** Functional summary of plant GELPs.

Crop Species	Reported GELPs	Process of Development	Reference
* **Arabidopsis thaliana** *	CDEF1	Lateral root development (auxin signaling-mediated); Pollen-stigma interaction and pollinator attraction; Lipid metabolism	[23]
	CUS2	Flower development	[31]
	DAD1	Flower development; Anther and pollen development	[29]
	EXL4	Pollen-stigma interaction and pollinator attraction; Lipid metabolism	[43]
	EXL6	Anther and pollen development	[6,38]
	GELP15, GELP39	Flower development	[32,33]
	GELP22, GELP38, GELP49, GELP51 and GELP96 GELP12, DELP55, GELP72, GELP73 and GELP81	Lateral root development (auxin signaling-mediated)	[21]
	GELP77	Anther and pollen development	[37]
	GER1	Coleoptile elongation; Lateral root development (auxin signaling-mediated)	[20,22]
	GLIP1	Plant-biotic stress interaction	[62]
	GLIP2	Lipid metabolism; Plant-biotic stress interaction	[22]
	LIP1	Seed germination	[19]
	OSP1	Plant-abiotic stress interaction; Stomata development	[28]
	PLIP2, PLIP3	Plant-abiotic stress interaction	[61]
	RVMS	Anther and pollen development	[36]
	SFAR4	Lipid metabolism; Plant-abiotic stress interaction	[51]
	WDL1	Plant height; Stomata development	[26]
* **Brassica** * * **napus** *	BnSCE3, BnLIP2	Seed germination; Seed development	[18]
* **Capsicum** * * **annuum** *	CaGLIP1	Plant-abiotic/biotic stress interaction; Lipid metabolism	[50]
* **Glycine max** *	GmGELP28	Plant-abiotic stress interaction	[9]
* **Gossypium** * * **hirsutum** *	GhGDSL	Cell wall metabolism	[56]
	GhGLIP	Seed development	[48]
* **Jacaranda** * * **mimosifolia** *	JNP1	Pollen-stigma interaction and pollinator attraction; Lipid metabolism	[44]
* **Oraza sativa** *	BS1	Cell wall metabolism	[55]
	DARX1	Cell wall metabolism	[54]
	DR	Plant height	[26]
	MHZ11	Root development	[24]
	OSGELP34, OsGELP110, OsGELP115	Anther and pollen development	[40,41]
	OsGLIP1, OsGLIP2	Plant-biotic stress interaction; Lipid metabolism	[52]
	RMS2	Anther and pollen development; Lipid metabolism; Secondary metabolism	[39]
	WDL1	Plant-abiotic stress interaction	[27]
* **Solanum** * * **lycopersicum** *	CD1	Secondary metabolism; Flower development; Fruit development; Plant-abiotic stress interaction	[14,30,47,57]
	CUS1	Flower development; Fruit development	[14,30]
	GDSL1	Fruit development; Plant-abiotic stress interaction; Secondary metabolism	[46]
	SlCGT	Secondary metabolism	[59]
	SlGDSL2/SlCD1	Fruit development; Plant-abiotic stress interaction	[47,57]
	Sl-LIP8	Pollen-stigma interaction and pollinator attraction; Lipid metabolism	[45]
* **Tanacetum** * * **cinerariifolium** *	TcGLIP	Secondary metabolism	[58]
* **Zea mays** *	IPE2	Anther and pollen development	[42]
	ZmMs30	Anther and pollen development; Lipid metabolism	[8]

## Data Availability

Data is contained within the article.

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
