# Peer review of "Plant GDSL Esterases/Lipases: Evolutionary, Physiological and Molecular Functions in Plant Development"

_plants, 2022, doi:10.3390/plants11040468_

Round 1

Reviewer 1 Report

The review is clear, comprehensive and of relevance to the field. Spelling and grammar should be checked.

check the order of citation numbering, (citation numbers in Tables, or position of Tables should be reorganized)

Orazy sativa - Oryza sativa

Author Response

Responses to the comments from Reviwer #1.

Comment 1: The review is clear, comprehensive and of relevance to the field.

Response: Thanks for your supportive comments.

Comment 2: Spelling and grammar should be checked.

Response: We have checked all through the text and corrected spelling and grammar errors with the help of a native English speaker . All corrections have been blue colored.

Comment 3: check the order of citation numbering, (citation numbers in Tables, or position of Tables should be reorganized)

Response: Thank you for pointing out this mistake. We have checked the citation order as suggested.

Comment 4: Orazy sativa - Oryza sativa

Response: We have corrected it as suggested (Table 2 ).

Reviewer 2 Report

The manuscript by Shen et al., is of great interest for the roles of conserved GDSL esterases/lipases in plants. However, I have some concerns that should be addressed.

1) The authors started out with nice introduction on GDSL esterases/lipases. However, the introduction is lacking details for a review article. For example, line 34 “Unlike other lipolytic enzymes, GELPs have flexible active sites, broadly diverse sub- 34 strates and multifunctional properties”. This should be more elaborated. Perhaps substrates of other lipolytics enzymes and GDSL esterases/lipases could be compared.

2) line 38: please expand on “secure food supply under changing climate conditions”

3) line 65: “With the advances in genomics, particularly comparative genomic studies, combined with the molecular understanding of the process of cuticle formation and interactions between cuticle and other macromolecules, it is conceivable that exact roles of GELPs in plant land colonization will be elucidated eventually.” This statement is somewhat of a stretch.

4) The writing style and of this manuscript will greatly benefit from some extensive editing of English.

5) Line 207: lipases like cutinase = lipases-like cutinase

6) Line 405: “Combined with reverse genetics using genome editing, biochemical 405 and molecular characterizations, exact roles of GELPs in plant development and stress 406 response will be ultimately elucidated”. Again, this sentence is a little bit of a stretch.

7) Line 407: “which will facilitate crop breeding to secure food 407 supply in a sustainable way.”. Please elaborate.

Author Response

Responses to comments from Reviewer #2

General comment: The manuscript by Shen et al., is of great interest for the roles of conserved GDSL esterases/lipases in plants. However, I have some concerns that should be addressed.

Response: Thank you for providing constructive comments to improve our MS’ quality.

Comment 1: The authors started out with nice introduction on GDSL esterases/lipases. However, the introduction is lacking details for a review article. For example, line 34 “Unlike other lipolytic enzymes, GELPs have flexible active sites, broadly diverse substrates and multifunctional properties”. This should be more elaborated. Perhaps substrates of other lipolytics enzymes and GDSL esterases/lipases could be compared.

Response: This is a nice suggestion. We have added elaborated description to compare GDSL/ esterases/lipases with other lipolytic enzymes and relevant references (lines 33-38).

Comment 2: line 38: please expand on “secure food supply under changing climate conditions”

Response: We have revised this sentence as suggested (Lines 41-43).

Comment 3: line 65: “With the advances in genomics, particularly comparative genomic studies, combined with the molecular understanding of the process of cuticle formation and interactions between cuticle and other macromolecules, it is conceivable that exact roles of GELPs in plant land colonization will be elucidated eventually.” This statement is somewhat of a stretch.

Response: We have revised it as suggested (Lines 66-67).

Comment 4: The writing style and of this manuscript will greatly benefit from some extensive editing of English.

Response: Thanks. We have asked helps from an English native speaker colleague to improve the writing quality (All corrections are blue colored).

Comment 5: Line 207: lipases like cutinase = lipases-like cutinase

Response: We have corrected as suggested (Line 210).

Comment 6: Line 405: “Combined with reverse genetics using genome editing, biochemical and molecular characterizations, exact roles of GELPs in plant development and stress response will be ultimately elucidated”. Again, this sentence is a little bit of a stretch.

Response: We have revised it as suggested (Lines 409-411).

Comment 7: Line 407: “which will facilitate crop breeding to secure food supply in a sustainable way.”. Please elaborate.

Response: We have revised it as suggested (Lines 410-411).

Round 2

Reviewer 2 Report

Thank you for making edits as suggested. I believe that the current version of this manuscript warrants publication in Plants.

This manuscript is a resubmission of an earlier submission. The following is a list of the peer review reports and author responses from that submission.

Round 1

Reviewer 1 Report

“Plant GDSL esterases/lipases: evolutionary, physiological and moelcular functions in plant developmentsby Gaodian Shen et al.

The authors have summarized the present knowledge concerning GDSL esterases/lipases of several plants including A thaliana. Even though they have summed up, but ms is very weak to publish. After reading this ms, my opinion is that this ms is not sufficient in this status. Therefore, I think that it is not acceptable.

Reviewer 2 Report

Dear authors

Thanks a lot for your manuscript on GDSL lipases. I do really like your goal to provide an overview on recent literature on these enzymes. However, in the current shape it contains too many unclear sections/sentences to have sufficient impact or to bring your message across. Please take your time to improve the manuscript, the English language and the visualization/Tables of the data you want to include. That will enhance the message and will make everything much more clear for potential readers. In the attached .pdf document I have made some suggestions on the initial pages, but similar corrections are valid throughout the manuscript. I wish you good luck and am looking forward to reading a revised version of the manuscript! 
